# Protein structure determination in human cells by in-cell NMR and a reporter system to optimize protein delivery or transexpression

Juan A. Gerez [1✉], Natalia C. Prymaczok[1], Harindranath Kadavath[1], Dhiman Ghosh[1], Matthias Bütikofer[1], Yanick Fleischmann [1], Peter Güntert [1,2,3] & Roland Riek [1✉]

Most experimental methods for structural biology proceed in vitro and therefore the contribution of the intracellular environment on protein structure and dynamics is absent. Studying proteins at atomic resolution in living mammalian cells has been elusive due to the lack of methodologies. In-cell nuclear magnetic resonance spectroscopy (in-cell NMR) is an emerging technique with the power to do so. Here, we improved current methods of in-cell NMR by the development of a reporter system that allows monitoring the delivery of exogenous proteins into mammalian cells, a process that we called here "transexpression". The reporter system was used to develop an efficient protocol for in-cell NMR which enables spectral acquisition with higher quality for both disordered and folded proteins. With this method, the 3D atomic resolution structure of the model protein GB1 in human cells was determined with a backbone root-mean-square deviation (RMSD) of 1.1 Å.

[1] Laboratory of Physical Chemistry, ETH Zürich, 8093 Zürich, Switzerland. [2] Institute of Biophysical Chemistry, Goethe University Frankfurt, Max-von-Laue-Str. 9, 60438 Frankfurt am Main, Germany. [3] Department of Chemistry, Tokyo Metropolitan University, 1-1 Minami-Osawa, Hachioji, 192-0397 Tokyo, Japan. ✉email: juan.gerez@phys.chem.ethz.ch; roland.riek@phys.chem.ethz.ch

Altthough the cellular environment can affect the structure, function, and reactivity of biomolecules, the current atomic-resolution experimental methods for the structural biology of proteins proceed in vitro, and hence the information related to the complex network of interactions that these molecules undergo in the intracellular milieu is absent. Studying protein structure and dynamics at atomic resolution in living mammalian cells has been challenging due to the lack of methodologies. In an effort to overcome this limitation, in-cell NMR has emerged as a powerful technique to analyze macromolecules inside living cells with atomic resolution[1,2]. However, despite its unique potential in structural biology, progress in in-cell NMR has been slow[3,4], and only a handful of proteins could be studied by in-cell NMR to date[5,6]. This is mainly attributed to the observation that in-cell NMR spectra feature low signal intensities. Several factors contribute to the low quality of in-cell NMR spectra. The most critical ones are (i) the suboptimal quantities of NMR-visible protein of interest present in the cells, and (ii) the multiple interactions between the protein under study and molecules of the cell interior. While (i) constitutes a technical challenge that can be addressed by the development of new methods and protocols, (ii) strongly depends on specific and relatively long-lived high-affinity interactions as well as transient low-affinity interactions that take place inside the cells[7]. These interactions lead to slower tumbling and chemical exchange of the protein under study, both resulting in line broadening of the NMR resonances[7].

In-cell NMR of mammalian systems requires that the protein of interest is isotopically labeled and present in living cells. This NMR-visible protein can either be expressed by the same cells that are subjected to the NMR determinations[8] or provided externally[9]. Despite its advantages, the endogenous expression of the isotopically labelled protein features an increased background of NMR signals due to the uncontrollable synthesis of other isotopically-labeled biomolecules that are generated in mammalian cells growing in culture media containing these isotopes. The signal resulting from these molecules is substantial. It may be partially subtracted from the spectra obtained with cells expressing the protein of interest[10]. Alternatively to endogenous expression, the isotopically labeled and NMR-visible protein can be obtained from an external source and then introduced into mammalian cells by microinjection (e.g. as in the case of frog oocytes)[11,12], electroporation[13], cell-penetrating peptides[9], or pore-forming toxins[14]. As the mammalian cells are not exposed to isotopically-labeled precursors, the main advantage of these methods is a reduced background of the NMR signals. A second advantage is the possibility of monitoring immediate structural changes occurring on proteins lacking any modification or eventually modified in a specific manner according to the aim of the experiment[15–17]. Regardless of the method, having optimal amounts of the isotopically-labeled protein inside the cells is critical for obtaining good-quality spectra by in-cell NMR. In this work, we improved current methods of in-cell NMR based on the delivery of exogenous proteins into mammalian cells. We first developed a reporter system for the fast and reliable quantification of protein delivery into mammalian cells. The reporter mimics the initial steps required to carry out in-cell NMR as it relies on an exogenous protein that is introduced into mammalian cells, an approach introduced here as "transexpression". With this reporter we optimized a method for in-cell NMR which allowed us to obtain higher quality spectra of multiple proteins as well as to determine for the first time the atomic resolution structure of a folded model protein with a backbone RMSD of 1.1 Å in living mammalian cells.

## Results

**"Transexpression" or delivering proteins into mammalian cells.** In contrast to RNA and DNA, which have been delivered into mammalian cells at different amounts by standardized methods over the last decades, to date, the delivery of exogenous proteins into mammalian cells has not been studied in detail neither mechanistically nor quantitatively. Because the process of introducing proteins into mammalian cells has not been semantically defined yet, and as first step towards its standardization and improvement, here we termed it "transexpression". Transexpression refers therefore to the introduction of an "exogenous" protein into mammalian cells by any of the different experimental methods for such aim (Supplementary Fig. 1a). The method can be electroporation, cell penetrating peptides, cell membrane permeabilization by toxins, lipoparticles, or any other experimental method used to deliver proteins into mammalian cells. For "exogenous protein" it must be stated that the delivered protein necessarily needs to be obtained from an external source, that in most cases is an heterologous system. Upon intracellular delivery, the exogenous protein is now "transexpressed" in the recipient cell, forming part of its own proteome. As recipient cells do not express the exogenous protein but instead it is artificially acquired, the transexpression term fulfills the semantic requirements for such experimental method.

**Development of a reporter system to evaluate protein delivery into mammalian cells termed "transexpression".** Aiming at establishing standardized methods for the controlled and efficient transexpression, we first developed a reporter system to quantitatively study how proteins can be introduced into mammalian cells. The system has two components; (i) a functional protein that can be efficiently expressed and purified in heterologous systems, and (ii) a reporter gene for quantifying the activity of the protein selected in (i) (Fig. 1a). We chose the chimeric transcription factor Gal4-VP16 (Sadowski et al., 1988) as the protein target because it has several favorable properties: it has been extensively used in mammalian cells, its activity can be easily assayed in vivo and in cell lysates, it has a molecular weight suitable for solution state NMR (~19 kDa), a globular 3D structure as most mammalian proteins, and is absent in mammals precluding evolutionary conserved interactions within the mammalian cell that might interfere with the readouts used to evaluate delivery efficiency[18]. For (ii) we developed a DNA vector containing a reporter gene called pGal4-5XRE-eGFP, in which the mutant gene of the *Aequorea victoria* enhanced green fluorescent protein (eGFP) was cloned downstream of a minimal promoter containing five GAL4 binding sites or responsive elements (RE) (Fig. 1a). The reporter system works as follows: first pGal4-5XRE-eGFP is stably or transiently introduced in mammalian cells by standard transfection methods. In a second step, the recombinant protein Gal4-VP16 (rGal4-VP16) is delivered into these cells by any of the different methods of transexpression. As eGFP expression is induced by the exogenously added rGal4-VP16, the transexpression efficiency positively correlates with eGFP fluorescence signal intensity in these cells.

As proof of principle of our reporter system, we first investigated whether eGFP could be induced by transexpressing rGal4-VP16 into mammalian Cos7 cells. To this end, rGal4-VP16 was expressed and purified from bacteria, and different amounts thereof were delivered into cells that had been previously transfected with pGal4-5XRE-eGFP. The transexpression method in this first experiment was electroporation; rGal-VP16 was first mixed with a cell pellet and then an electric pulse was applied to the mixture. After electroporation, the Cos7 cells were washed twice with fresh media in order to remove rGal4-VP16 that had not been internalized. The cells were then plated, grown at 37 °C for 24 h, harvested, and the obtained cell suspensions were subjected to eGFP signal intensity quantification. We found a

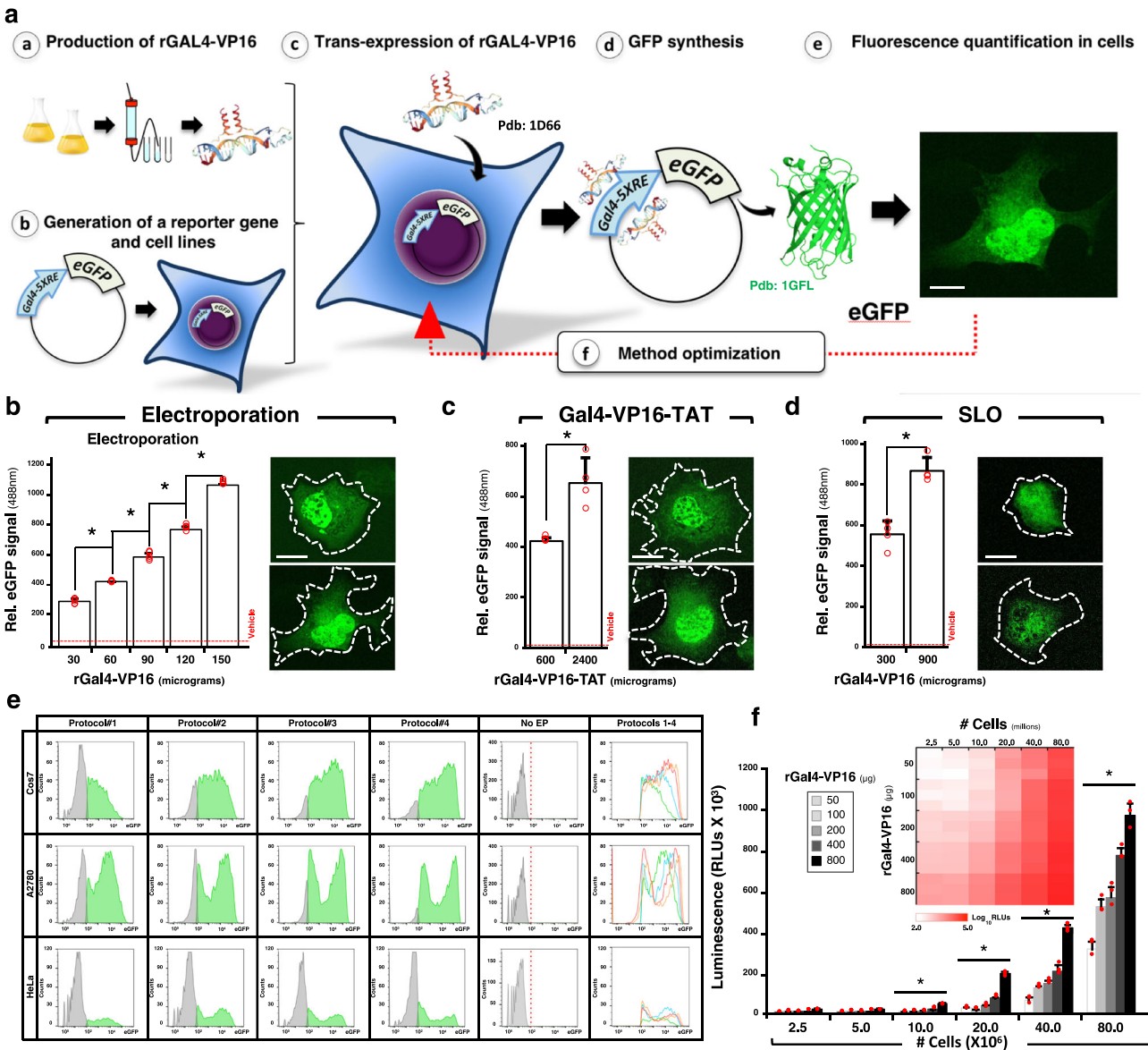

**Fig. 1 A reporter system for "transexpression". a** Schematic representation of the reporter system based on recombinant Gal4-VP16 (rGal4-VP16) and the artificial gene Gal4-5XRE-eGFP. rGal4-VP16 is produced and purified from bacteria (**a**), and mammalian cells stably transfected with pGal4-5XRE-eGFP are generated (**b**). Next rGal4-VP16 is transexpressed into these cells (e.g. by electroporation, cell penetrating peptides or pore-forming toxins) (**c**). The delivered rGal4-VP16 induces eGFP expression in these cells (**d**), whose fluorescence emission at 488 nm can be easily quantified (**e**). By testing different experimental conditions, transexpression is optimized and the desired amount of rGal4-VP16 delivered to the cells is obtained (**f**). These experimental conditions can now be used to transexpress the protein of interest. Gal4 (bound to DNA) and eGFP structures were taken from pdb: 1D66 and 1GFL. Scale bar 10 μm. **b–d** Transexpression of different amounts of rGal4-VP16 in Cos7 cells transfected with pGal4-5XRE-eGFP ($n = 3$ biologically independent samples). The transexpression methods were electroporation (**b**), the cell penetrating peptide TAT (**c**), and the pore-forming toxin streptolysin-O (SLO) (**d**). The values of cells treated with vehicle is shown by dashed red lines. Confocal images of the transexpressed cells are shown on the right. The results are expressed as means + SD. *$p < 0.05$ [one-way analysis of variance (ANOVA), followed by Dunnett's post hoc test]. Scale bar 10 μm. **e** Four different protocols differing on the buffer composition were used to transexpress by electroporation rGal4-VP16 into Cos7, A2780 and HeLa cells. Fluorescence was quantified by FACs. Fluorescence of non-electroporated cells (No EP) and the overlap of protocols 1-4 is shown on the right. **f** Protein amount and cell number dependence on transexpression by electroporation. Increasing amounts of 2.5 to 80 million cells were used to transexpress increasing amounts of rGal4-VP16 (from 50 to 800 μg) in cells previously transfected with the plasmid pG5-Luc which encodes the firefly luciferase whose expression is under the control of Gal4-VP16 ($n = 3$ biologically independent samples). The bar plot shows the luminescence values of cells (in millions) electroporated with the different amounts of rGal4-VP16, while the heat map also shows the values obtained with different cell number. The region of the bar plot corresponding to 2.5 to 10 million cells is magnified in Supplementary Fig. 1d. The results are expressed as means + SD. *$p < 0.05$ [one-way analysis of variance (ANOVA), followed by Dunnett's post hoc test] compared to lower number of cells.

substantial increase of eGFP-derived fluorescence in Cos7 cells electroporated with rGal4-VP16 (Fig. 1b). The signal intensity was proportionally higher when increasing amounts of the transcription factor were used for electroporation, validating the

Gal4 system for the quantitative determination of protein delivery efficiency. The dynamic range in this experiment was from 30 to 150 μg and no saturation of rGal4-VP16 activity was observed within this range. Very low fluorescence levels were observed

when only buffer was used in the electroporation step, presumably due to cell autofluorescence and uncontrolled expression of pGal4-5XRE-eGFP in these cells. Confocal microscopy analyses confirmed the intracellular and robust induction of eGFP in cells electroporated with rGal4-VP16 (Fig. 1b).

We next assayed whether this reporter system can be used to evaluate other transexpression methods such as those based on cell-penetrating peptides[9] and cell membrane permeabilization by pore-forming toxins[14]. To this end, we fused the TAT HIV cell-penetrating peptide to the C-terminus of rGal4-VP16[9]. The resulting protein, rGal4-VP16-TAT, was then produced in bacteria and used to treat Cos7 cells previously transfected with pGal4-5XRE-eGFP. We found that eGFP was induced in cells treated with this chimeric transcription factor but not with vehicle (Fig. 1c). Compared to the electroporation-based method, the signal intensity obtained was significantly lower and about one order of magnitude more recombinant transcription factor was needed to obtain the values reached with the electroporation procedure. The intracellular localization of eGFP was also confirmed in this experiment (Fig. 1c). Likewise, eGFP fluorescence was observed in pGal4-5XRE-eGFP-transfected Cos7 cells treated with the pore-forming toxin streptolysin-O (SLO) and rGal4-VP16, whereas only basal levels of fluorescence were found in cells treated with the toxin and vehicle only (Fig. 1d). The fluorescence obtained with the SLO method was around a factor of five lower when compared to electroporation, and higher amounts of rGal4-VP16 were required in this experiment. Altered cell morphology (Fig. 1d) and considerable cell death was observed when using SLO, presumably due to its intrinsic cytotoxic effect on mammalian cells.

Compared to the TAT and SLO-based methods, a more efficient intracellular delivery of rGal4-VP16 was thus obtained with electroporation. Electroporation does neither require fusion peptidic tags such as TAT that might affect protein structure and function, nor toxins such as SLO that might trigger cell responses aimed to counteract cytotoxicity. Moreover, the delivery of proteins by electroporation is fast allowing the immediate analysis of the delivered protein by NMR. Because of these major advantages, for subsequent experiments we decided to concentrate on electroporation as the transexpression method.

The reporter system was next challenged on three different cell lines (Cos7, A2780, and HeLa) using different protocols of electroporation, of which four are shown in Fig. 1e. These four protocols differed only in the composition of the buffer used in the electroporation step. In this case, we analyzed eGFP signal intensity by fluorescent activated cell sorting (FACS), as this methodology allowed us to visualize the number of cells expressing eGFP as well as the signal intensity distribution. We found that among the cell lines assayed A2780 and Cos7 are most suitable for rGal4-VP16 transexpression using these electroporation parameters (Fig. 1e and Supplementary Fig. 1b, c). Protocol 1 resulted in Cos7 cells with relatively low eGFP expression, while the majority of the Cos7 cells subjected to protocols 3 and 4 expressed high amounts of this protein. When using protocol 2, relatively similar numbers of Cos7 cells expressing all possible levels of eGFP were found. Surprisingly, eGFP signal intensity showed two peaks in A2780 cells, indicating that two populations of cells were obtained; cells with moderate eGFP levels and cells with high levels of this protein. The population corresponding to cells expressing high amounts of eGFP was favored with protocol 1. The electroporation conditions used in this experiment were less efficient for HeLa cells, which showed only low eGFP expression. Changing the electroporation parameters (e.g., pulse shape, length and voltage), however, allowed us to achieve efficient transexpression also for HeLa cells. Thus, we conclude that the reporter based on Gal4-VP16 is a valuable tool to find the optimal experimental conditions that allow us to deliver the desired amount of a given protein into mammalian cells.

Towards establishing a standardized method of transexpression for in-cell NMR studies, we next investigated how the amount of rGal4-VP16 and the number of cells used in the electroporation step can affect transexpression efficiency. To this end, we electroporated different amounts of rGal4-VP16 in samples containing increasing quantities of Cos7 cells. In this experiment the cells were previously transfected with the plasmid pG5-Luc (Promega), which contains five binding sites for Gal4 to drive the expression of the firefly luciferase (Luc) by rGal4-VP16. The advantage of using luciferase is that its activity can be determined in cell lysates and not in intact cells as it was done for eGFP signal intensity determinations (Fig. 1b–e). In agreement with the eGFP values shown in Fig. 1b, we found that luciferase transcriptional activity correlated positively with both the amount of rGal4-VP16 and the number of cells used for electroporation (Fig. 1f and Supplementary Fig. 1d). Interestingly, the relationship between transcriptional activity and protein amount and cell number was in all cases positive and appears to be steeper than linear. Because the contribution of the cell number on transexpression efficiency showed more than a positive linear response and appears to be of complex nature, we then investigated the impact of cell size on transexpression efficiency by comparing the intracellular delivery of rGAL4-VP16 in four lines of cells with different sizes, Cos7 and U2OS with an average size of 30–40 μm, and the two smaller cell types Hek-293 and A2780 with an average size of 10–15 μm (Supplementary Fig. 1e). Based on the requirements of in-cell NMR (see Fig. 2), after electroporation, the cells were collected and packed into glass tubes of 3 or 5 mm diameter till they filled the "NMR active" region of the tubes. This height was approximately 20 millimeters for the cryoprobe of our NMR spectrometers (Supplementary Fig. 1f). Packing the cells was carried out by a gentle centrifugation step (300 × g for 2 min) that preserves >95% of cell viability (see Fig. 3). As these cells are of different sizes, the number of Hek-293 and A2780 cells packed in these two fixed sample volumes was higher than for Cos7 and U2OS cells. In 5 mm tubes, for example, it was >2 times higher (Supplementary Fig. 1g). We found that at a fixed sample volume smaller cells displayed higher levels of luciferase activity compared to Cos7 and U2OS cells (Supplementary Fig. 1h). This is presumably due to the increased surface-to-volume ratio of smaller cells that would be favorable for protein entry by electroporation. Thus, small cells are a better option when methods with limited sample size (volume) are used such as NMR. These differences are lost, however, when normalization by the cell number is applied to luciferase activity levels.

**Development of a transexpression method for highly sensitive in-cell NMR**. The reporter system based on rGal4-VP16 allowed us to establish a highly efficient transexpression method for the analysis of proteins by in-cell NMR. The method consists in protein delivery by electroporation, followed by a washing step of the electroporated cells for the efficient removal of the non-internalized protein, a recovery phase, in which cells are re-plated and dead cells are discarded, and a final step where the cells are harvested and packed into the NMR tube (Fig. 2a).

Aiming at carrying out long (>16 h) in-cell NMR experiments, we investigated first how the temperature and the incubation time affect the viability of packed cells. To this end, mock-electroporated Hek-293 and A2780 cells were packed, incubated at different temperatures, and finally recovered at different time points to determine cell viability using the trypan blue exclusion test. We found that ∼ 95% of the cells that stayed in the tube for < 8 h were viable, regardless of the temperature used (Fig. 2b). For the two cell lines assayed, at 16 h and later time points, only cells incubated at 37 °C showed a small but significant reduction in cell

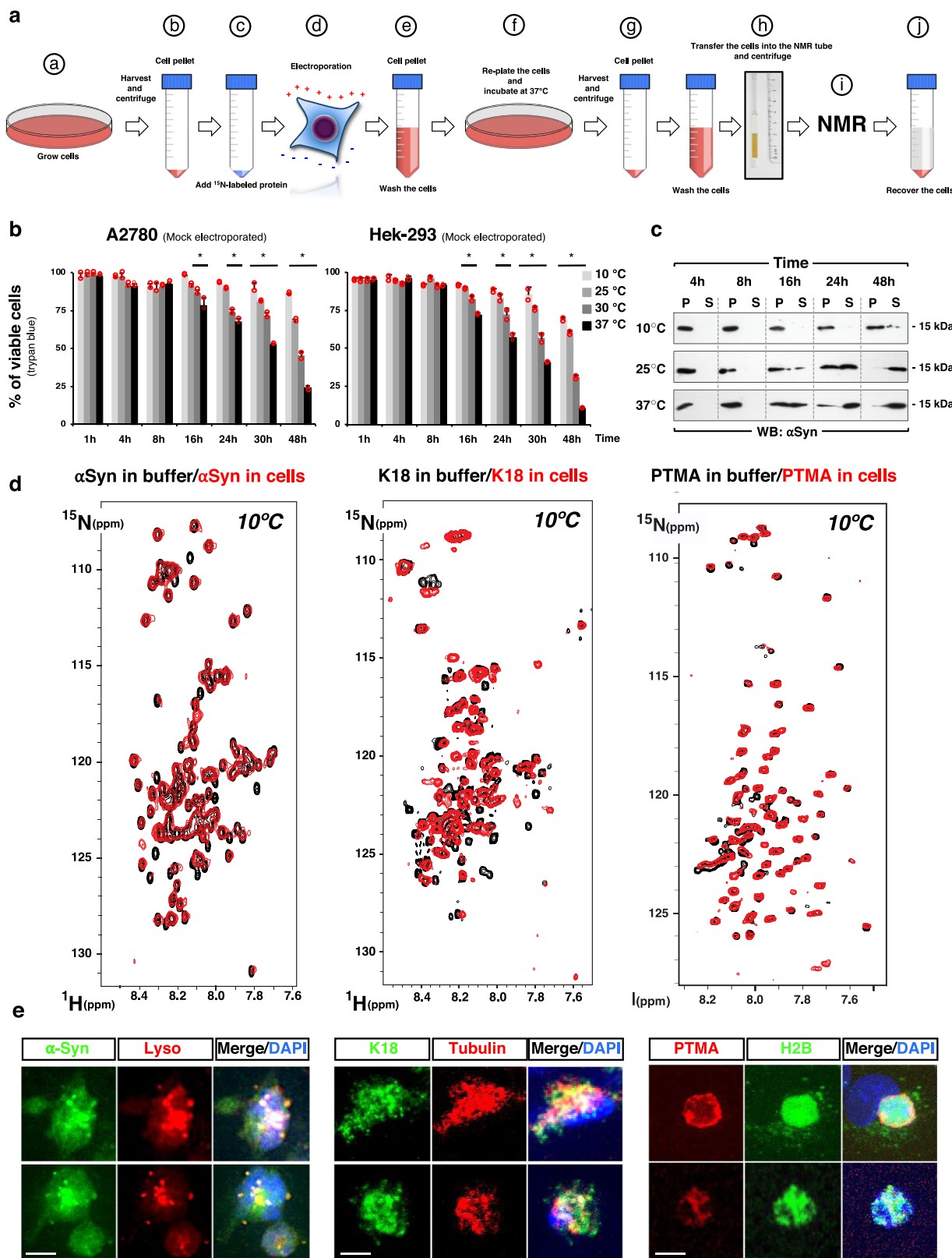

viability, whereas after 24 h this effect was also observed in cells kept at 30 °C. For longer times, high viability was only observed at 10 °C and 25 °C, where ~ 80% and ~ 70% of the cells remained alive at 24 and 48 h, respectively.

We next analyzed the loss of the plasma membrane integrity, an important parameter for in-cell NMR experiments, as plasma membrane leakage might result in the release of the transexpressed

protein to the media during the NMR measurements. Using the mammalian protein α-synuclein (αSyn), which was delivered using the transexpression protocol described above, we confirmed that at 10 °C the majority of the electroporated protein remained in the cells even 48 h after electroporation (Fig. 2c and Supplementary Fig. 2c). In agreement with the aforementioned trypan blue test, higher temperatures led to a faster release of this protein into the

**Fig. 2 An improved method of in-cell NMR. a** Schematic representation of the method established for efficient transexpression and in-cell NMR analyses. The desired number of cells are grown in culture dishes at a confluence of 90% (**a**), then harvested and centrifuged (**b**), and the resulting cell pellet is mixed with a solution containing the protein under study that must be visible by NMR e.g. must be isotopically labeled with [15]Nitrogen (**c**). An electroporation pulse is applied to the mixture of cell suspension and protein using the parameters found with the reporter system (**d**). Next, the non-internalized protein is removed by centrifugation and three washes with fresh media are applied (**e**). The cells are then re-plated and grown at 37 °C (**f**). After 2-4 hours the cells are harvested again and washed twice with fresh media (**g**). The cells are centrifuged and the pellet is transferred into an NMR tube, which is also centrifuged in order to obtain a cell pellet in the tube (packed cells) (**h**). The in-cell NMR measurements usually are > 1 hour of duration (**i**). After these determinations the cells are recovered from the tube (**j**) and further analyzed (e.g. cell viability, presence of the [15]N-labeled protein in the supernatant and pellet). **b** Cell viability of mock-electroporated A2780 (left) or Hek-293 (right) cells ($n = 2$ biologically independent samples) that were packed in the NMR tube and incubated at different temperatures for the indicated times. The results are expressed as means + SD. * $p < 0.05$ [one-way analysis of variance (ANOVA), followed by Dunnett's post hoc test]. **c**) Cells transexpressed with the protein alpha-synuclein (αSyn) were packed into the NMR tube and incubated at 10, 25 and 37 °C. after 4, 8, 16, 24 and 48 hours the cell membrane integrity was assayed by detecting by Western blot αSyn in the pellet (P) or supernatant (S) of the packed cells. **d** In-cell NMR of IDPs. Two dimensional [[15]N,[1]H]-Heteronuclear Multiple Quantum Coherence (HMQC) NMR spectra of [15]N-labeled αSyn, K18 Tau and prothymosin-α (PTMA) transexpressed in mammalian cells (red spectra). The reference spectra correspond to the same proteins in buffer (black spectra). NMR measurements were carried out at 10 °C. **e** Confocal microscopy analyses of Cos7 cells transexpressed with αSyn, K18 Tau and PTMA. Co-staining for lysosomes, histone 2B (H2B) and β3-tubulin was included. Scale bar 10 μm.

extracellular media. Thus, the data indicated that whereas at 10 °C high cell viability is obtained for short and long incubation times, caution has to be taken when higher temperatures and incubation times longer than 16 h are used. In this sense, incubation at 37 °C is limited to ≤ 8 h.

We next carried out in-cell NMR experiments on three proteins using the protocol described in Fig. 2a. We included recently published protocols[19,20] for comparative purposes. The main differences between the protocol described in Fig. 2a and the previously published ones are the amounts of cells and recombinant protein needed for electroporation, the electroporation buffers, the electroporation parameters and the electroporation devices (see Materials and Methods). In this comparative experiment, however, the amounts of cells and recombinant protein were the same for the previously published protocol and the method described in Fig. 2a. The proteins of choice were the intrinsically disordered proteins (IDPs) αSyn, prothymosin-α (PTMA) and K18, a fragment of the tau protein[21,22]. IDPs were selected for initial experiments because their fast tumbling usually results in sharp, strong NMR signals. These three proteins were produced and purified from bacteria as [15]N-labeled proteins, delivered into mammalian cells by the two electroporation protocols, and measured by two dimensional [[15]N,[1]H]-HMQC NMR experiments. Compared to the previously published protocols, our optimized protocol yielded on average 3 times higher signal-to-noise ratios (Fig. 2d compared to Supplementary Fig. 2b, Supplementary Fig. 2a and c). We confirmed this by comparing two electroporation buffers and devices in two cell lines (Supplementary Fig. 2d). While the Nucleofector IIb (AMAXA) yields good transexpression efficiency with the buffer "R", the Neon (Invitrogen) is superior when PBS is used. These differences might be due to the fact that the two devices use different electroporation units (cuvettes versus tips). Thus, we conclude that a higher efficiency of transexpression is obtained with the new protocol.

The NMR spectra also revealed that PTMA and K18 remain disordered inside mammalian cells (Fig. 2d) as previously shown for αSyn[13]. Compared to the spectra obtained with the protein in buffer (Fig. 2d and Supplementary Fig. 2e, black), there was line broadening, signal attenuation, and chemical shift perturbation in the in-cell [[15]N,[1]H]-HMQC spectra of the three proteins. For instance, quantifying peak intensities of the spectra obtained in cells and in buffer revealed several regions of αSyn with peak attenuations in the in-cell spectrum, in addition to previously reported N-terminal acetylation[23,24]. In αSyn, the N- and C-termini as well as the region around tyrosine 39 are strongly affected by the intracellular milieu (Supplementary Fig. 2f). These changes were recently attributed to transient interactions with cellular partners such as chaperones[20]. Likewise, the spectra of K18 tau displayed alterations in several NMR resonances in the C-terminal regions of exons 1-2, and in exon 3 (Supplementary Fig. 2f). These effects might be due to interaction of K18 with microtubules and lipids[21,22,25]. In agreement with previous work[20], the electroporated αSyn was found in association with lysosomes in normal cells, as shown here by its co-localization with the lysosome-specific dye Lysotracker (Fig. 2e). Likewise, K18 is co-localized with microtubules[25] as shown by double immunofluorescence using anti-tau and anti-β3-tubulin antibodies, and PTMA was found to be localized in the nucleus and associated to histone 2B as previously reported[26] (Fig. 2e). The data altogether indicated that the electroporated proteins reach different destinations within the cell where they might play functional roles. Next, we analyzed the structural stability of the electroporated αSyn over time by carrying out in-cell NMR experiments at 10 °C, where cell survival and protein stability were maximal (Fig. 2b and Supplementary Fig. 2g, h). No significant chemical shift changes were observed over 16 h while some signal decay was detected. The data indicated that the IDP structure of αSyn remains unaltered in the NMR tube for at least 16 h (Supplementary Fig. 2e, g, h), allowing long NMR measurements under these experimental conditions.

**In-cell NMR of folded proteins at physiological conditions.** Anecdotal observations indicated that transexpression fails often for folded proteins. To counter this problem, we used our improved protocol of transexpression to carry out in-cell NMR experiments on five different folded proteins. We selected the β1 immunoglobulin binding domain (GB1) and the third IgG-binding domain from streptococcal protein G (GB3), the PDZ2 domain of the human tyrosine phosphatase 1E (PDZ), phosphoglycerate kinase 1 (PGK1), and wild type ubiquitin (Ub) because they represent a broad spectrum of folded proteins that includes non-mammalian proteins (GB1 and GB3), proteins with enzymatic activity (PGK1), a protein-protein interaction domain (PDZ), and a protein involved in signal transduction and post-translational modification of proteins (ubiquitin). Aiming at the analysis of proteins in physiological conditions, most experiments were conducted at 37 °C in relatively short NMR experiments. PDZ in contrast was analyzed at 25 °C because the effect of the intracellular milieu on this protein at 37 °C was within the NMR time scale (< 1 h) immediate. With the transexpression protocol shown in Fig. 2a, we were able to obtain high-quality spectra for all five proteins, as shown in Fig. 3a. Overall, all five proteins show a similar in-cell [[15]N,[1]H]-HMQC spectrum as the in vitro

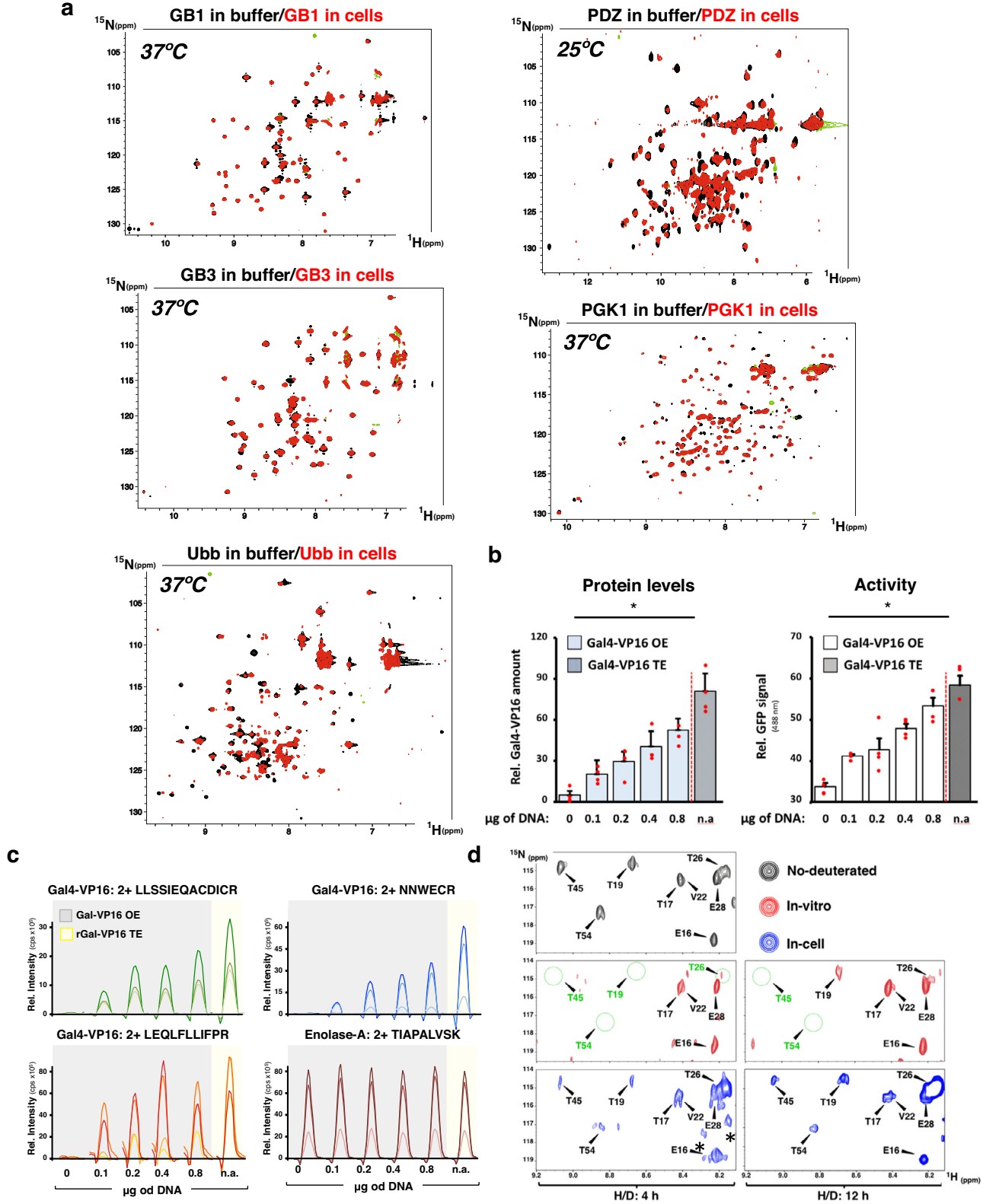

reference (Fig. 3a). Since the [$^{15}$N,$^1$H]-HMQC spectrum is a fingerprint of the protein structure it can be concluded that all five proteins have the same overall 3D structure in cells as in vitro. However, compared to the in vitro reference, substantial line broadening, signal attenuation, and chemical shift perturbations of cross-peaks were observed in the [$^{15}$N,$^1$H]-HMQC spectra for the five proteins. Compared to the three IDPs

analyzed above, the peak intensity alterations were much more pronounced and widespread in the folded proteins, as shown in Fig. 3a. In all cases the two spectra (in buffer and in cells) show similar cross peak linewidths because the in-cell samples contain significantly more amount of protein than the one in buffer and the signal-to-noise was adjusted to illustrate the superposition of the spectra. At similar protein concentrations the in-cell spectra

**Fig. 3 In-cell NMR of folded proteins. a** In-cell NMR of folded proteins. Two dimensional [$^{15}$N,$^{1}$H]-HMQC NMR experiments of $^{15}$N-labeled GB1, PDZ2 domain of protein human tyrosine phosphatase 1E, GB3, PGK1 and wild type ubiquitin transexpressed in mammalian cells (red spectra). The reference spectra (black, overlapped) corresponds to the same proteins in buffer. NMR measurements were carried out at 37 °C except for the PDZ2 domain which was measured at 25 °C. **b** Activity tests for the ectopically-expressed (OE, from overexpression) Gal4-VP16 and transexpressed (TE, from transexpression) rGal4-VP16 in Hek-293 cells ($n = 4$ biologically independent samples). Hek-293 cells stably transfected with pGal4-5XRE-eGFP were transiently transfected with different amounts (0 to 0.8 μg) of an expression plasmid encoding Gal4-VP16. 48 h later the cells were harvested and Gal4-VP16 protein levels as well as activity (eGFP signal intensity) were determined. In parallel, rGal4-VP16 was transexpressed in these cells and included in these analyses. The bar plots show the relative levels (on the left) and activity (on the right) of Gal4-VP16. The results are expressed as means + SD. *$p < 0.05$ [one-way analysis of variance (ANOVA), followed by Dunnett's post hoc test]. **c** Quantification of intracellular levels of ectopically-expressed (OE, from overexpression) and electroporated (TE, from transexpressed) Gal4-VP16 in Hek-293 cells by parallel reaction monitoring mass spectrometry. Increasing amounts of Gal4-VP16 were expressed by these cells, which were then lysed and subjected to mass spectrometry analyses. The housekeeping gene enolase-A was used for loading control. As surrogates for intracellular protein levels, three tryptic peptides of Gal4-VP16 and one of enolase-A were quantified. Targeted peptides are shown at the top of each plot, and at least four transitions of the y-series of the product ions were monitored over the chromatographic separation of the peptides (different colors). cps, counts per second. **d** Hydrogen-deuterium (H/D) experiments on GB1 in buffer and in cells. The exchangeable protons of $^{15}$N-labeled GB1 were exchanged by deuterium and then this protein was transexpressed into mammalian cells. Inside the cells as well as in buffer exchangeable protons of GB1 exchanges back to protons, which are now visible in the [$^{15}$N,$^{1}$H]-HMQC spectrum. With time signal enhancement is indicated. NMR measurements were carried out at 4 and 12 hours post electroporation. As reference in vitro PBS buffer was used. To account for signal loss during incubation time in cells the 12 h spectrum shown was normalized by adopting the contour lines of the cross peaks of the fast exchanging residues E16, T17 and E28 to the corresponding cross peaks in the 4 h spectrum. Green circles, resonances of slow exchange. * indicates noise.

show substantial line broadening compared to the one in buffer, as shown for GB3 in Supplementary Fig. 3a.

The [$^{15}$N,$^{1}$H]-HMQC fingerprints of all eight proteins studied suggest functional integrity of the transexpressed proteins. It could however be, that a significant amount of transexpressed protein is deteriorated in the electroporation step and the NMR spectra collected originate from a small remaining soluble and functional fraction of the total delivered protein. We addressed this question by quantifying how much of the delivered protein is functional. To this end, the protein levels and transcriptional activity of the transexpressed rGAL4-VP16 was determined, and compared to a GAL-VP16 that is produced by the same host cells and therefore fully functional. We first generated Cos7 cell clones stably transfected with pGal4-5XRE-eGFP. These cells were then either electroporated with rGAL4-VP16 or transiently transfected with a mammalian expression vector encoding GAL4-VP16. Both cells were finally harvested and the eGFP signal intensity as well as the protein levels of both the transexpressed rGAL4-VP16 and the episomally-expressed GAL4-VP16 were both determined in whole cell lysates. Protein quantification was carried out by a targeted proteomics approach called Parallel Reaction Monitoring (PRM)-mass spectrometry[27,28], which was used to quantify three different peptides of this chimeric transcription factor as surrogate for the total amount of this protein in these cells. We found a linear correlation between the amount of endogenously expressed GAL4-VP16 and eGFP signal intensity, confirming that the vast majority of the protein in the cells is functional (Fig. 3b, c). Importantly, the protein levels and transcriptional activity of the transexpressed rGAL4-VP16 fit into the curves of the transcription factor produced by the cells, indicating that the electroporated protein is also functional.

We next used hydrogen-deuterium (H/D) exchange experiments to investigate whether the transexpression process unfolds the electroporated protein. To this end, GB1 was expressed in bacteria as $^{15}$N-labeled protein, then unfolded and incubated in $D_2O$ buffer to replace the naturally occurring exchangeable $^{1}$H protons by deuterium followed by refolding. Transexpression into mammalian cells by electroporation using a buffer prepared with PBS salts dissolved in deuterium was performed followed by in-cell NMR experiments 4 and 12 h thereafter with the same sample recording the H/D exchange. If the protein unfolds during electroporation, all amide deuterons are expected to exchange fully back to protons before the NMR measurements requesting equivalent relative signal intensities both at 4 and 12 h after taking

into account protein loss with time. In buffer and in cells, after 4 h most resonances of deuterated GB1 were already visible, with some of them at the levels of non-deuterated GB1. This indicated a fast H/D exchange for the latter residues, which comprised threonine 17 (T17), glutamate 28 (E28), and glutamate 16 (E16), among others (Fig. 3d and S3b and S3c). Of note, all these residues are solvent-exposed and thus expected to exchange fast. In contrast, some residues, such as threonines 19, 26, 45, and 54 (T19, T26, T45 and T54, respectively) displayed amides that had not exchanged fully neither in vitro nor in cells. These findings show that GB1 did not unfold during electroporation and thus was incorporated into the cells as a folded protein answering the critical question whether transexpression by electroporation may harm the integrity of the protein structure to be delivered. As expected for longer incubation times, after 12 h some of these amide moieties showed increased exchange (Fig. 3d). Interestingly, H/D exchange appeared to be faster in cells than in vitro. For instance, the intensities of the resonances corresponding to T19, T26, T45 and T54 were lower in vitro than in cells (Fig. 3d and Supplementary Fig. 3b, c). This is more evident upon normalization to the cross peaks of the fast-exchanging signals (i.e. E16, T17 and E28) correcting for time-related signal loss in cells at 12 h (Supplementary Fig. 3c). These data indicate that the protein structure is less stable in cells than in vitro, a phenomenon previously reported for ubiquitin[9]. This could result from chaperone interactions with GB1 to be explored[9].

**Structure determination in mammalian cells by NMR.** Next, we wanted to test whether also a structure determination by in-cell NMR is feasible in our system of mammalian cell lines as documented for insect cells[29] and in E. coli[30]. For this, GB1 with its outstanding spectral properties was selected and transexpressed in Hek-293 cells. These cells contained electroporated $^{13}$C,$^{15}$N-labeled GB1 with a final concentration of 50 μM, estimated by comparison to a reference in vitro sample (Supplementary Fig. 4a). A 3D [$^{15}$N,$^{13}$C]-combined [$^{1}$H,$^{1}$H]-NOESY experiment with a mixing time of 200 milliseconds was measured for 1 day at 10 °C for the collection of the NOE-derived distance restraints. The low temperature was used to preserve cell integrity (Fig. 2). Time-dependent spectral changes were minor as indicated by a comparison of the [$^{15}$N,$^{1}$H]-HMQC spectra at various time points (Supplementary Fig. 4b). In addition, a 3D [$^{15}$N,$^{13}$C]-combined [$^{1}$H,$^{1}$H]-NOESY of $^{13}$C,$^{15}$N-labeled GB1 at a concentration of 0.5 mM was measured in PBS buffer (pH 7.4) with the same setup. The sequential assignment

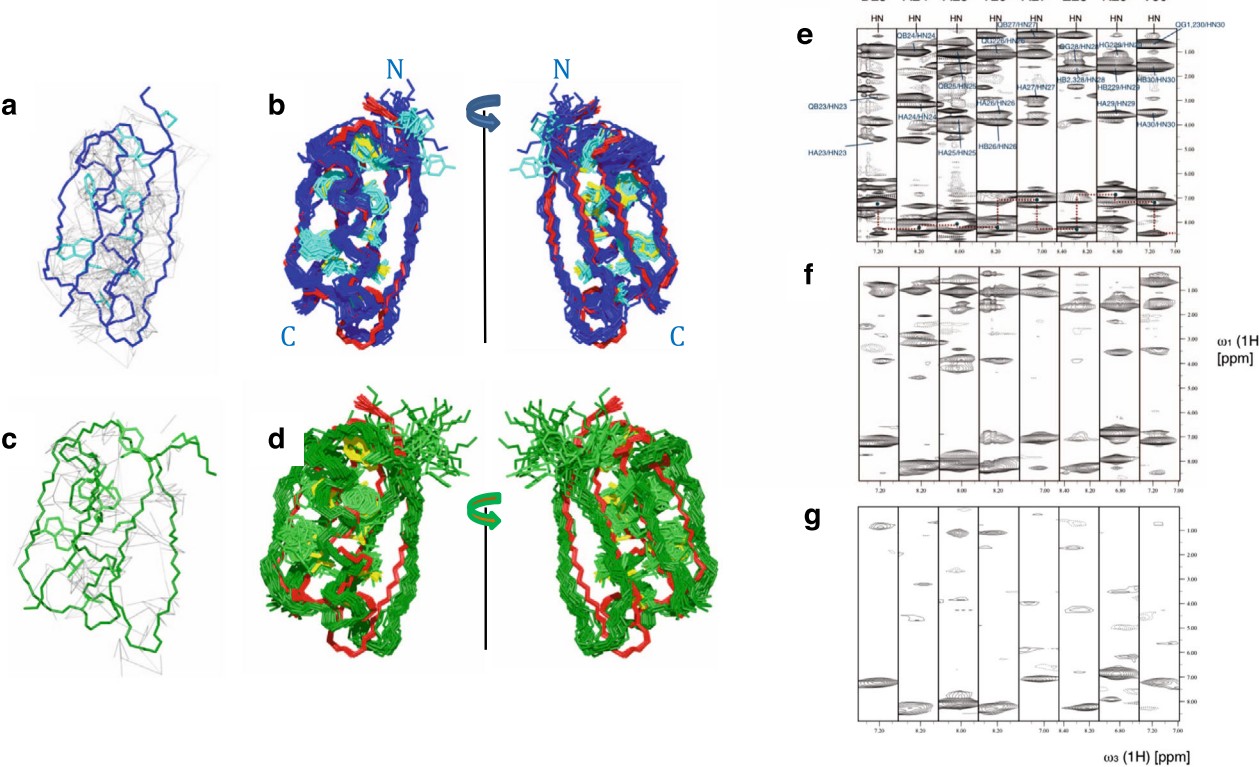

**Fig. 4 Structure determination in mammalian cells. a**, **b** The 3D NMR structures of GB1 in cells at a protein concentration of 50 μM and (**c**, **d**) at a concentration of 10 μM. In **a** and **c** the NOE-derived distance restrains are shown in grey on top of a representative conformer of the calculations. The backbone of the structure is shown in blue and green, while the most buried side chains are colored more light (i.e. cyan or light green). In **b**, **d** the two calculated structures are shown in the bundle representation comprising 20 conformers highlighting the RMSD of the structure calculation. The same coloring as in **a**, **c** is used. In addition, the 3D in vitro NMR structure with the PDB code 2N9K.pdb is superimposed to the in-cell structures and color coded in red for the backbone and in yellow for the buried side chains. Again 20 conformers are shown representing the 3D structure quality.

**e-g** $^{15}$N,$^{13}$C-resolved [$^1$H,$^1$H]-NOESY spectra of **e** reference GB1 in vitro, **f** in cell at 50 μM concentration and **g** in cell at 10 μM. The HN/N strips of residues D23-V30 are shown as indicated. In **e** the intra residual assignment is depicted in blue, the diagonal peaks are indicated by a blue circle, and the anticipated sequential walk from HN to HN is indicated by red dotted lines. It is evident that a significant signal loss is observed from **e** to **f** and a further vast signal loss is observed from **f** to **g**.

as well as distance restraint collection of the latter positive control was obtained starting from the available chemical shift list[30]. We transferred the NOESY assignment (including the sequential assignment) to the NOESY spectrum measured in cells. This included minor adjustments on the chemical shifts as well as the loss of many NOE cross peaks due to the 10-fold lower concentration, yielding 592 meaningful NOE-derived distance restraints (Fig. 4a), which are 678 distance restraints less than detected in the in vitro control sample. The loss of cross peaks attributed to the lower concentration is manifest in a comparison of Fig. 4e with 4f, which show the $^{15}$N-$^1$H strips of residues D23-V30 in vitro and in cells, respectively. Nevertheless, these NOE-derived distances were sufficient to determine the 3D structure (PDB 7QTR, BMRB 34700) with a backbone RMSD to the mean of 0.9 Å for residues 2–57 and an RMSD of 1.1 Å to the reference structure (Fig. 4a and Supplementary Table 1, PDB 2N9K). Also, the 25% most buried side chains superimpose well with the published in vitro structure as shown in Fig. 4a–d. Only around the N-terminus the structure does not superimpose well with the reference structure and is less well defined, which is attributed to the low number of distance restraints in this area (Fig. 4a–d). The CYANA target function, which is a measure of experimental distance restraint violations[31] has a small value of 1.44 Å$^2$, indicating that the experimental data are self-consistent (Supplementary Table 1).

It is evident that a concentration of 50 μM is well above the natural concentration of most mammalian proteins. Towards a

more physiological situation, we prepared a set of 4 identical samples, each one containing 10 μM $^{13}$C,$^{15}$N-labeled GB1 transexpressed in Hek-293 cells (Supplementary Fig. 4a). For each sample, we measured the same NOESY experiment with a duration of 1 day at 10 °C. The individual spectra were summed up and the same analysis as stated above was performed. This yielded 187 distance restraints as manifest in Fig. 4g. The structure calculation was only successful if using also torsion angle restraints derived by TALOS-N[32] from the secondary chemical shifts that could also be extracted from the [$^{15}$N,$^{13}$C]-combined [$^1$H,$^1$H]-NOESY spectrum. The structure is of low quality but still of atomic resolution with an RMSD of 1.5 Å to the mean and 2.2 Å to the reference structure (Supplementary Table 1; PDB 7QTS, BMRB 34701). A comparison of this in-cell structure with the reference structure manifests that it is less compact, which is attributed to the low number of restraints that hold the structure less together than requested. In summary, for GB1 at near physiological concentration, a 3D structure at atomic resolution with an accuracy of ca. 2 Å was determined, which shows the correct fold as well as the side chain arrangement of the buried core residues.

## Discussion

Introducing exogenous molecules into mammalian cells constitutes a very common experimental strategy in the field of cell and molecular biology and alike. As the exogenously added

molecule can display functionality, introducing these molecules into cells has been instrumental for the analysis and comprehensive understanding of a myriad of biological processes. While a small molecule such as certain metabolites and drugs can be introduced into cells spontaneously by its administration to the culture media and subsequent uptake, other molecules with a less favorable chemical nature or increased size need assistance to be internalized. This is the case for nucleic acids such as RNA and DNA, which have been delivered intracellularly at different amounts by standardized methods such as transfection or viral transduction over the last decades. In contrast to RNA and DNA, the efficient delivery of exogenous proteins into mammalian cells has not been studied in detail neither mechanistically nor quantitatively and hence no unifying method has been established. Because successful only in few particular cases, efficient delivery of proteins into mammalian cells represents a major limitation and technical challenge for the development of experimental methods aimed to analyze proteins that the cells cannot produce properly and in adequate amounts. No terminology has been provided for this technique to date, and therefore here we named "transexpression" to the process of delivering exogenous proteins into mammalian cells.

To date, studying protein structure and dynamics at atomic resolution in living cells has been successful only in a small number of cases[33,34]. Since its introduction more than 20 years ago[3,4], in-cell NMR was supposed to fill this gap but progress has been slow[5,6]. In-cell NMR features low signal-to-noise ratios that render the spectra suboptimal for such analyses. Much effort has been undertaken over the last years to improve this technique, including the development of more sensitive NMR methods[35,36] a refinement of the cell culture conditions to maximize isotope labelling[37–40], and the development of bioreactor systems for the continuous supply of fresh medium into the NMR tube[41,42], which allows long NMR measurements, e.g. 3D NOESY spectra for collecting distance restraints. Also, paramagnetic tags have been used to measure pseudocontact shifts (PCSs) and residual dipolar couplings (RDCs)[43,44]. On the contrary, substantial improvements related to protein delivery have not been reported to date. Because efficient transexpression represents a major limitation and technical challenge for the experimental methods aimed to analyze proteins that the cells cannot produce properly or in adequate amounts (i.e., isotopically-labeled proteins), we first developed a reporter system to easily monitor and optimize transexpression. The reporter system is based on the intracellular delivery of an exogenous protein with biological activity such as a chimeric transcription factor Gal4-VP16. This reporter has several advantages which are critical for optimizing transexpression; first, the transexpressed rGal4-VP16 needs to reach the cell nucleus to activate the reporter gene for eGFP. This implies that the reporter system can provide information on whether or not the transexpressed protein is able to reach different destinations within the cell such as the nucleus or organelles. Second, in addition to protein delivery, the reporter also allows to test whether the transexpressed protein remains structurally intact upon delivery. This is inferred from functional studies using different amounts of transexpressed protein. Last, it is known that certain proteins have affinity for lipids and membranes, and that remain attached to the cells but from the extracellular domain upon transexpression. We have seen this behavior when transexpressing recombinant eGFP as an alternative method to evaluate transexpression. However, we found that a substantial amount of eGFP remains attached to the cells even when the electroporation pulse is intentionally not applied. The reporter developed here is devoid of this bias.

We successfully applied the reporter in transexpression methods based on electroporation, cell-penetrating peptides, and reversible permeabilization of the plasma membrane by SLO. As both the Gal4 and VP16 fragments that are comprised in the rGal4-VP16 chimera are endowed of a well-defined 3D structure[45–47], it is expected that the intracellular delivery of the resulting fusion protein is mechanistically similar to the delivery of most small-sized (10 to 40 kDa) globular proteins. But the conditions of transexpression that we found to be optimal for delivering rGal4-VP16 yielded also highly efficient transexpression of the three intrinsically disordered proteins studied in this work, and the fact that rGal4-VP16 and αSyn, K18 tau and PTMA are structurally very different suggests this reporter could also be used to find the experimental conditions optimal for transexpressing proteins of different characteristics. Ongoing experiments using alternative exogenous transcription factors will shed light into the actual limitation of rGal4-VP16 as screening tool to improve transexpression of proteins of different nature.

In addition to find the best conditions for efficient transexpression, the reporter system has also other applications and some of them were described in this work; with this tool we demonstrated that, using these specific experimental conditions, transexpression by electroporation might be superior to cell-penetrating peptides or SLO. The reporter also revealed that smaller cells yield higher transexpression than larger cells, a critical factor for in-cell NMR because the number of cells that can be packed into the NMR-sensitive probe volume is limited. We found in certain cell lines, such as A2780, the presence of different cell populations, each containing different amounts of the transexpressed protein. Future studies aimed at separating these two populations (e.g. by cell sorting) will unravel how the structure and dynamics of a protein behave at different stoichiometric ratios of intracellular partners.

The reporter was instrumental to carry out functional studies on the transexpressed protein; the experiments with rGAL4-VP16 showed that the electrical pulses applied for electroporation did not functionally or structurally damage the transexpressed protein. That the transexpressed protein is functional is in line with spectral indications that the tau isoform K18 interacts with microtubules, and the observation that αSyn is N-terminally acetylated upon transexpression as in cells[23,24] both suggest that transexpression via electroporation delivers a functional protein. Co-localization of αSyn with lysosomes[20], K18 with β3-tubulin[48], and PTMA with histone 2H[26] supports the idea of functionality of the transexpressed proteins. However, it is known that proteins have different susceptibilities and some of them are prone to undergo irreversible changes that lead to loss-of-function. Therefore, we anticipate that there should be proteins that behave different to rGal4-VP16 and therefore are damaged upon electroporation. In this sense, transexpression of rGal4-VP16 could still be useful as it is possible to use stronger electroporation parameters that damage rGal4-VP16. These experimental conditions can thus be used when screening for conditions where the deleterious effect of electroporation on the transexpressed protein is minimized. Last but not least, it is important to note that although the reporter system based on rGal4-VP16 was used here to improve in-cell NMR, its use extends beyond structural biology since it can be used also for optimizing functional protein delivery as for example for stem cell differentiation, genome editing and alike[49].

Aiming at releasing the potential of in-cell NMR for in vivo structural biology[1,2], the methods developed in this work were applied to eight proteins of varying structure, function, origin, and size, culminating in the structure determination of the model protein GB1 in living human cells. To date only a handful of protein structures have been solved in living cells, and most of these structures were obtained from *E. Coli* expressing the protein of interest at high levels. Indeed, very few protein structures have

been obtained from eukaryotic cells. Likewise, in most of the structures solved in bacteria and eukaryotic cells, NMR was combined with additional methods such as PCSs or PRE to obtain structural information from well-resolved in-cell distance restrains. To date, the most commonly used eukaryotic system for structure determination is *Xenopous laevis* oocytes, which allowed to solve the structure of GB1 by NMR and PCSs/PRE[43,44]. In an outstanding work, Tanaka and colleagues solved the structure of five proteins living insect sf9 cells[29], demonstrating that de novo protein structure determination using NMR exclusively was possible in eukaryotes cells. Structure determination in mammalian cells seems to be more challenging and to date only one protein structure has been solved in these cells[50]. In previous works the structure of a ubiquitin triple mutant was solved in HeLa cells by using PCSs. Thus, to our knowledge, the GB1 structure reported here is the first protein structure solved entirely by in-cell NMR in mammalian cells, without relying on PCSs, PRE or structure prediction software. The method is therefore suitable to analyze proteins that cannot be mutated or modified with paramagnetic tags.

We summed four identical NOESY experiments to determine the structure of GB1 at nearly physiological conditions that include an intracellular concentration of 10 μM. The number of experiments that can be combined to improve the NOE distance restraint collection is, in principle, unlimited, making it possible to reach even lower concentrations and/or shorter life times of proteins or cells. It is important to mention that these NMR measurements were carried out at 10 °C and as at this temperature both cell leakage and cell viability was preserved for 24 h presumably due to lower metabolic state of the cells compared to 37 °C. Ongoing experiments involving bioreactors to preserve cell viability in longer NMR experiments carried out at 37 °C will shed light onto how feasible is to determine the structure of other folded proteins at physiological conditions using the presented method. As reported for the triple mutant ubiquitin[50], for the 3D structure determination of future systems, the NOE distance restraint collection as used here could be complemented by other in-cell NMR data, such as PCSs and PREs[43,44], as well as by non-linear sampling NMR experiments aimed to reduce the acquisition time of 3D spectra[51].

It is important to note that although improvements on trans-expression described here were used to solve the structure of a robust protein model (GB1) in mammalian cells using in-cell NMR, the higher sensitivity and resolution of the obtained spectra will allow more detailed mechanistic and functional atomic-resolution analyses in mammalian cells of proteins that at the present lack of spectra of good quality. The presented method represents a first approach to those structures that are greatly affected by the cell interior and therefore cannot be compared to the spectra obtained in vitro[52]. In summary, this work establishes in-cell NMR as a promising tool for structure-activity relationship studies, the elucidation of dynamics, and structure determination at atomic resolution for proteins or protein domains residing inside mammalian cells.

## Methods

**Plasmids and cloning**. Plasmid construction and DNA manipulations were performed following standard protocols. The DNA sequences of all constructs were verified by sequencing prior to use. The bacterial expression plasmid encoding rGAL4-VP16 (pRJR-Gal4VP16) was provided by Stephen Buratowski (Harvard Medical School, USA). This plasmid encodes a 6×HIS-tagged GAL4-VP16 fusion protein consisting of the DNA binding domain of the yeast Gal4 protein (amino acids 1–93) fused to the C-terminal 78 amino acids of the activation domain of the VP16 protein of the herpes simplex virus (Sadowski et al., 1988). The rGAL4-VP16-TAT-encoding plasmid (called pET11a-GAL4(1-93)-VP16-TAT) was obtained by fusing the cDNA coding region of the HIV TAT protein in frame with the 3'-end of the coding region of pRJR-Gal4VP16. pET11a-GAL4(1-93)-VP16-TAT was generated by PCR using the following primers:

Gal4VP16UppHIS: 5'ATATA<u>CATATG</u>CACCATCACCATCACCACAAGCTACTGTCTTCTATCGAACAAGC '3;

Gal4VP16Low: 5'GGAATTGACGAGTACGGTGGGTATGGACGCAAGAAGAGGAGGCAAAGAAGGAGATAGTAGG<u>GGATCC</u>GGCTG '3;

The PCR product was then digested and ligated into the NdeI and BamHI (in bold) sites of pET11a. To express GAL4-VP16 in mammalian cells we used the plasmid pSCTGal (1-93)-VP16 (kindly provided by Prof. Walter Schaffner and Dr. Oleg Georgiev, University of Zurich, Switzerland), which encodes the DNA binding domain of the yeast Gal4 protein (amino acids 1–93) fused to the C-terminal 80 amino acids of the activation domain of the VP16 protein of the herpes simplex virus (Sadowski et al., 1988).

To generate pGAL4-5XRE-eGFP, the CMV promoter of the plasmid pEGFP-N1 (Clontech) was replaced by the minimal promoter of the pG5-Luc plasmid (Promega), which contains five binding sites for Gal4 (called 5XRE) and the major late promoter of adenovirus. The minimal promoter was amplified by PCR using the following primers:

TB/Gal4 upp: 5' TACG<u>ATTAAT</u>ATGCATCTTGGAGCGGCC '3;

LucNrev: 5' CCTTATG<u>CAG</u>TTGCTCTCC '3;

digested with AseI (in bold) and HindIII, and ligated to pEGFP-N1 previously digested with the same restriction enzymes.

We generated a bacterial expression plasmid encoding non-tagged PTMA by removing the DNA sequence encoding a 6×HIS-tag from the pET47b-PTα vector (kindly provided by Prof. Benjamin Schuler, University of Zurich, Switzerland). The coding region of PTMA was amplified from pET47b-PTα using the following primers:

PTa-HISless_upp: 5' ATC<u>GACATATG</u>TCAGACGCAGCCGTAGACACC '3;

STag_18mer_Rev: 5' GTCCA<u>TGTGCT</u>GGCGTTC '3;

digested with NdeI (in bold) and SacI, and ligated into pET47b digested with the same restriction enzymes. The bacterial expression plasmid of αSyn was pRK172-αSyn. The plasmid for bacterial expression of wild-type Ubiquitin was provided by Prof. Matthias Peter (ETH Zurich, Switzerland). The plasmid encoding K18 tau was kindly provided by Prof. Marc Diamond (UT Southwestern, USA). The plasmid encoding PGK1 is in the pET28a(+) backbone and was purchased from GenScript. The plasmid encoding GB1 was kindly provided by Angela Gronenborn (University of Pittsburgh, USA).

**Cell culture and transfections**. Cells were cultured in Dulbecco's modified Eagle's medium (DMEM) (pH 7.4) supplemented with 10% fetal calf serum (FCS), 4 mM L-glutamine, 100 U/ml penicillin, and 100 mg/ml streptomycin. Transfection assays were performed with lipofectamine-2000 reagent (Invitrogen) following the manufacturer instructions.

**Production of recombinant proteins**. Unless stated otherwise, all recombinant proteins were produced in *E. coli* BL21 Star (DE3) cells as [15]N or [15]N-[13]C-uniformly labeled proteins as follows: a pre-culture of 200 mL was prepared with LB media and grown overnight at 37 °C in agitation (120–180 rpm). Then 20 ml of the preculture was added to each one of six flasks containing 2 L (1:100 dilution) of 2XLB media (20 g/L NaCl, 20 g/L tryptone and 10 g/L yeast extract). These cultures were grown at 37 °C to OD600 = 1, the cells were then collected by centrifugation (4000 × *g* for 15 min), and finally transferred into three flasks containing each 2 L of minimal media containing the isotopes. After 2 h IPTG (1 mM final concentration) was added to the cells, which were incubated overnight at 37 °C in agitation. Cells were harvested by centrifugation and stored as cell pellets at −80 °C until they were purified.

rGal4-VP16 and rGal4-VP16-TAT were produced as non-labeled proteins. The corresponding plasmids were transformed into BL21-DE3 cells, which were grown in LB medium at 37 C to OD600 = 0.5, and then treated with 1 mM IPTG. Before IPTG was added, the media was supplemented with 20 μM ZnSO4. The cells were grown for additional 4 h at room temperature and then harvested. Cells were lysed by sonication at 4 °C in lysis buffer (10 mM TrisCl pH: 8, 500 mM NaCl, 10% glycerol, 10 mM β-mercaptoethanol, 0.1% Tween-20, 10 μM ZnSO4, 10 mM imidazole, 0.01% Triton X-100 and 1 mM PMSF), and clarified by centrifugation. The supernatant was incubated over night at 4 °C with Ni-NTA resin, and then packed into a column. One wash of four column volumes was applied (20 mM HEPES-KOH pH: 7.6, 100 mM NaCl, 20% glycerol, 1 mM DTT, 1 mM EDTA, 10 μM ZnSO4, 20 mM Imidazole, and 1 mM PMSF), and eluted with a linear gradient of the same washing buffer plus 500 mM imidazole. Fractions containing rGal4-VP16 and rGal4-VP16-TAT were pooled and snap frozen in liquid nitrogen. For long-term storage the samples were stored at −80C. Recombinant [15]N-labeled αSyn was expressed by co-transformation of the pRK172-αSyn plasmid with the plasmid pNatB3 coding for the *S. pombe* NatB acetyltransferase complex[20]. For non-tagged [15]N-labeled PTMA, the cell pellet was resuspended in TE buffer pre-heated at 80 °C (25 mM Tris/HCl, 1 mM EDTA, pH 8). The cell lysate was further boiled at 100 °C for 5 min and then incubated on ice for 2–3 h. The lysate was sonicated three times at 4 °C, passed through a microfluidizer, and cleared by centrifugation (20,000 × *g* for 30 min). An 80% saturation of ammonium sulfate was then applied to the supernatant (incubation at 4 °C for 2–3 h), which was then centrifuged and the supernatant extensively dialyzed against buffer TE. The solution was filtered and loaded onto a HiTrapTM Q FF 16/10 anion exchange chromatography column (GE healthcare) equilibrated with TE buffer. Elution was

carried out with a linear gradient of 0–500 mM NaCl. The fractions containing PTMA were pooled, dialyzed against water, and lyophilized till use. K18 tau was expressed as [15]N-labeled protein and purified as described previously[53,54]. GB1 was produced as [15]N and [15]N,[13]C-labeled protein and purified as follows: the cell pellet was resuspended in purification buffer pre-heated for 10 min at 85 °C (10 mM Tris/HCl, 1 mM EDTA, pH 7.5). Three sonication steps were applied, and the lysis was completed by passing the cells three times through a microfluidizer. The resulting suspension was cleared by centrifugation (28,000 × $g$ at 4 °C for 30 min), and ammonium sulfate was added to the supernatant until reaching 80% saturation. The solution was incubated overnight at 4 °C with constant agitation, centrifuged at 15,000 × $g$ for 1 h, and the supernatant exhaustively dialyzed against 10 mM Tris-HCl, pH 7.5. EDTA was added to a final concentration of 1 mM, and the pH was adjusted to 7.5 with Tris/HCl. The protein solution was loaded onto an anion-exchange column (GE Helathcare HiPrep Q FF 16/10), eluted in a 0–500 mM NaCl gradient, dialyzed overnight against 10 mM TrisHCl, pH 7.5, and lyophilized till use. To produce [15]N-labeled PDZ domain, the cell pellet was resuspended at 4 °C in purification buffer (10 mM Tris/HCl, 200 mM NaCl, 10 mM imidazole, pH 8), then sonicated three times at 4 °C, then passed through a microfluidizer, and cleared by centrifugation (20.000 × $g$ for 1 h). the supernatant was filtered and loaded onto a nickel (II)-charged chelating sepharose FF column (Amersham Biosciences cat# 17-5255-01), equilibrated with purification buffer. The column was washed with 3 column volumes of the same buffer and eluted with a linear gradient of 10–250 mM imidazole. The fractions containing PDZ were pooled and concentrated till a sample volume of 200–300 µL. The sample was snap frozen and stored at –80 °C. Before use, an equal volume of 8 M urea solution was added to the protein solution and incubated for 10 min at room temperature. The sample was desalted using PD minitrap Sephadex G-25 columns (GE Healthcare).

For the expression of PGK1, the cells were grown at 37 °C in LB medium until an optical density of 0.8 was reached. The cells were centrifuged and the cell pellet was resuspended in [15]N-enriched minimal medium. After 1 h, protein expression was induced over night at 18 °C with 1 mM IPTG. The cells were resuspended in lysis buffer (50 mM TrisHCl, pH 7.9, 200 mM NaCl, 5 % glycerol, 10 mM imidazole) and lysed in a microfluidizer. The supernatant was loaded onto a nickel affinity column. After washing the column with 5 column-volumes of lysis buffer, the protein was eluted using a linear gradient of 10–500 mM imidazole. The fractions containing protein were dialyzed into an imidazole free buffer and the His tag was cleaved with TEV in 1:30 ratio overnight at 4 °C. TEV was removed with an additional Nickel affinity column and PGK1 was purified using size exclusion chromatography using NMR buffer (25 mM Tris, 50 mM NaCl, 2 mM TCEP and 0.02 % NaN₃).

GB3 was expressed as a uniformly [15]N-[13]C-labeled protein. Transformed cells were grown till they reached an OD of 1.0, when protein expression was induced with 0.2 mM IPTG, and further grown for 3 h at 37 °C. The cells were harvested by centrifugation and resuspended in 20 ml of PBS buffer per 1 L of culture. The cells were lysed in boiling water at 80 °C. The lysed cells were centrifuged and the supernatant was passed through a 0.22 µm syringe filter. The supernatant was loaded to a Superdex 75 HiLoad 26/60 (Amersham Biosciences) column, equilibrated with PBS pH 7.4. The fractions containing GB3 protein were collected and concentrated and buffer exchanged using a 2 kDa concentrator.

**Confocal microscopy.** Cos7 cells grown in 15 cm plates were subjected to transfection and electroporation (for transexpression) and then re-plated onto glass covers in 2 cm plates. The cells were washed three times with Mg-PBS buffer (10 mM magnesium in PBS), fixed with 4 % paraformaldehyde for 10 min, and permeabilized with 0.5% TritonX-100 for 10 min at room temperature, washed three times, and covered with coverslips with Vectashield Mounting medium (Vector Laboratories). Images were acquired with a laser-scanning confocal fluorescence microscope (Visitron spinning disk).

Antibodies used were anti-αSyn clone LB509 (Abcam), anti-PTMA ab247074 (Abcam), anti-β3 tubulin (D71G9) (Cell signaling) and anti-Tau 4-repeat isoform RD4, clone 1E1/A6 (Sigma Aldrich).

**Quantification of Gal4-VP16 by parallel reaction monitoring (PRM) mass spectrometry.** Cells were harvested in a buffer containing 8 M urea and 0.1 M NH₄HCO₃, and sonicated three times for 10 s with an ultrasonic probe device. Cell lysates (100 µg) were reduced with 12 mM dithiothreitol for 30 min at 32 °C and alkylated with 40 mM iodoacetamide for 45 min at 25 °C in the dark. Samples were diluted with 0.1 M NH₄HCO₃ to a final concentration of 2 M urea, and sequencing-grade porcine trypsin (Promega) was added to a final enzyme:substrate ratio of 1:100. Tryptic digestion was conducted at 32 °C for at least 16 h in the dark. The digestion was stopped by acidification to pH 3 with formic acid. The peptide mixtures were loaded onto Sep-Pak tC18 cartridges (Waters), desalted and eluted with 80% acetonitrile. All peptide samples were evaporated on a vacuum centrifuge to dryness, resolubilized in 0.1% formic acid, and immediately analysed by mass spectrometry. The peptide samples were analysed on a hybrid Quadrupole-Orbitrap mass spectrometer (Q-Exactive HF, Thermo Scientific) equipped with a Waters M-class UPLC system (Waters AG) operating in PRM acquisition mode. Three tryptic peptides of Gal4-VP16 (2+LLSSIEQACDICR, 2+NNWECR, and 2+LEQFLLIFPR), as well as one tryptic peptide of human enolase-A (2 + TIA-PALVSK) were quantified by PRM as surrogates for intracellular Gal4-VP16 and enolase-A protein levels, respectively. A standard curve was obtained by using increasing amounts of plasmidic DNA in the transfection reactions. For each target peptide at least three PRM transitions of the y-series of the resulting fragment ions were used for determination of peptide quantities. Dissolved samples were injected by a M-class UPLC system (Waters) operating in trap/elute mode. A Symmetry C18 trap column (5 µm, 180 µm x 20 mm (Waters)) and an HSS T3 C18 reverse-phase column (1.8 µm, 75 µm x 250 mm (Waters)) as separation column were used. The columns were equilibrated with 99% solvent A (0.1% formic acid (FA) in water) and 1% solvent B (0.1% FA in acetonitrile). Trapping of peptides was performed at 15 µL/min for 30 s and afterwards the peptides were eluted using a gradient of 1–40% B in 30 min and 40–98% B in 5 min with a constant flow rate of 0.3 µl/min at 50 °C. Data were analysed using Skyline (MacCoss, Version 3.7). Relative abundance of each peptide across different conditions is shown as the PRM transitions of each peptide visualized with Skyline. Results are expressed as counts per seconds of each PRM transition (at least three per peptide) over retention time.

**Cell viability.** A2780 and Hek-293 cells were grown in 15 cm culture dishes till they reached 90% of confluency. The cells were harvested, centrifuged at 300 × $g$ for 3 min, resuspended in PBS buffer, centrifuged again and resuspended in a small volume of PBS buffer pre-cooled at 4 °C. An equal amount of cell suspension was aliquoted into 5 mm Shigemi NMR tubes (seven in total), which were then centrifuged at 300 × $g$ for 3 min. The supernatant was removed from the cell pellets and the NMR tube plunger was inserted. Each tube was incubated at the indicated temperature for the indicated times, then the cells were recovered from the NMR tubes and resuspended in pre-warmed DMEM media containing 10 % FCS. Finally, the cells were stained with trypan blue dye following the manufacturer's instructions.

**Cell leakage.** Ten 15 cm plates containing 90% confluent A2780 cells were trans-expressed with [15]N-labeled recombinant αSyn using the method described in Fig. 2a. After recovery the cells were harvested, washed three times in PBS buffer, and aliquoted in 5 millimeters NMR Shigemi tubes (five in total), which were then centrifuged at 300 x $g$ for 3 min. The supernatant then was removed from the cell pellets and the NMR tube plunger was inserted. Each tube was incubated at the indicated temperatures for the indicated times before the cells as well as the supernatant were recovered, lysed in Laemmli buffer, and finally analyzed by Western blot.

**Previously published in-cell NMR method.** This method is described elsewhere[19,20], except for the amount of [15]N-labeled protein, which was reduced to match the experimental conditions used in the new method. In brief, up to 6 plates of 90% confluent Hek-293 and A2780 cells were used for one in-cell NMR experiment. These cells were grown in DMEM, pH 7.3 supplemented with 10% fetal calf serum, 4 mM L-glutamine, 100 U/ml penicillin and 100 mg/ml strepto-mycin for 2 days, then harvested, pooled, centrifuged at 200 × $g$ for 3 min, and kept at 37 °C for 10 min as cell pellets. A lyophilized aliquot of [15]N-labeled αSyn, PTMA or K18 tau containing 1 mg of protein was re-suspended in 200 µL of sterile PBS pH 7.4 (GIBCO) and sonicated for 5 min in a bath sonicator. The protein solution was used to gently resuspend the cell pellet. The cell suspension was mixed at a 1:1 ratio with electroporation buffer "R" (AMAXA-LONZA), and incubated at room temperature for 5 min. After that, 100 µL of the cell suspension was placed in an electroporation cuvette (AMAXA-LONZA), and one electric pulse was applied immediately to the samples with a Nucleofector-IIb electroporator (AMAXA-LONZA) using the program Q001. Immediately after electroporation, the cells were transferred to a 15 mL conical tube, and the process was repeated taking a second 100 µL aliquot of the non-electroporated cell suspension. All electroporated cells were collected in the same 15 mL conical tube and washed twice with pre-warmed media, plated, and incubated at 37 °C for 4 h. After this recovery period, the cells were harvested, washed twice with pre-warmed media, and once with PBS pH 7.4 containing 5% D₂O. The cells were finally transferred to a 5 mm Shigemi NMR tube and packed by soft centrifugation (300 × $g$ for 3 min).

**In-cell NMR method developed in this work (Fig. 2a).** Two plates of 95% confluent Hek-293 and A2780 cells were used for one in-cell NMR experiment. These cells were grown in DMEM, pH 7.3, supplemented with 10% fetal calf serum, 4 mM L-glutamine, 100 U/ml penicillin and 100 mg/ml streptomycin for 2 days, harvested, pooled, centrifuged at 200 × $g$ for 3 min, and kept at 37 °C for 10 min as cell pellets. A lyophilized aliquot of [15]N-labeled αSyn, PTMA or K18 tau containing 1 mg of protein was re-suspended in 400 µL of sterile PBS pH 7.4 (GIBCO) and sonicated for 5 min in a bath sonicator. These samples were used to resuspend the cell pellet, and the mixtures were incubated at RT for 5 minutes. After that, 100 µL of the cell suspension was taken using a Neon pipette tip (Invitrogen) and placed into the pipette holder filled with 4 mL of PBS and 0.5 mL of glycerol. One squared electroporation pulse of 20 milliseconds at 1400 V was applied to the sample using a NEON electroporator device (Invitrogen). The cell suspension was transferred to an Eppendorf tube and the process was repeated with a another 100 µL aliquot of the cell suspension till all electroporated cells were collected in the same Eppendorf tube. Next, a second electroporation pulse was applied following the same procedure. After this second electroporation step, the cells were collected in a 15 mL conical tube, washed twice with pre-warmed media, plated, and incubated at 37 °C for 4 h. After this recovery period, the cells were harvested, washed twice with pre-

warmed media, and once with PBS pH 7.4 containing 5% $D_2O$. The cells were finally transferred to a 5 mm Shigemi NMR tube and then packed by soft centrifugation ($300 \times g$ for 3 min).

**NMR measurements**. NMR experiments were performed on a Bruker 700 MHz and 900 MHz Avance III HD spectrometers equipped with a cryogenically cooled proton-optimized $^1H[^{13}C/^{15}N]$ TCI probe. Specifically, 2D $^{15}N-^1H$ SOFAST HMQC spectra (46) were acquired with a data size of $128 \times 512$ complex points for a sweep width (SW) of 28.0 ppm ($^{15}N$) and 16.7 ppm ($^1H$), 512 scans, 100 ms recycling delay (acquisition time ~ 4 h). NMR spectra were processed with either Topspin (Bruker) and Sparky (University of California, San Francisco) or PROSA[55] and CARA[56]. Visualization and data analysis were carried out in Sparky or CARA. NMR signal intensity ratios ($I/I_0$) were determined for each residue by extracting the maximal signal height of the cross-peaks from the respective 2D $^{15}N-^1H$ NMR spectra. 3D $^{15}N,^{13}C$-resolved $[^1H,^1H]$-NOESY experiments (48) were acquired on a 900 MHz spectrometer at 10 °C with a NOE mixing time $\tau_m = 200$ ms, $256 \times 55 \times 1{,}024$ complex points, sweep widths of 13.0 ppm ($^1H$), 35.2 ppm ($^{15}N$), and 14.3 ppm ($^1H$), and 8 scans (acquisition time ~1 d).

**H/D exchange measurements**. GB1 was expressed in bacteria as a $^{15}N$-labeled protein and purified as described above, then lyophilized and stored at –80 °C. The protein was resuspended and unfolded by incubation in 6 M guanidine hydrochloride containing $D_2O$ to replace the naturally occurring exchanging $^1H$ protons by $^2H$. The guanidine hydrochloride was then replaced by PBS buffer made with $D_2O$ (a tablet of Phosphate saline buffer from SIGMA (cat#P4417) was resuspended in 200 ml of $D_2O$) using a PD SpinTrap G-25 column (GE Healthcare). For the H/D experiments an aliquot of the protein was 1:50 diluted into standard PBS buffer (without $D_2O$) and measured by NMR after 4 and 12 h of incubation at 37 °C. Transexpression of deuterated GB1 into mammalian A2780 cells was performed and immediately after the protein was analyzed by in-cell NMR at 37 °C in experiments of 4 h and 12 h duration. The same sample was used to record the H/D exchange after 4 and 12 h by NMR. To account for signal loss with time due to signal loss in cells and to normalize for relative transexpressed protein amount when compared to the in vitro control, cross peak intensities of fast exchanging $^{15}N-^1H$ moieties identified in the in vitro control experiment (i.e. $^{15}N-^1H$ moieties of Glu17, Thr18, and Val23, Figure S3a and S3b) were used.

**Structure determination**. Structures of GB1 at 50 and 10 μM concentration in human Hek-293 cells were calculated on the basis of manually assigned NOE distance restraints with CYANA using standard parameters[31]. Chemical shift assignments were adapted from a previous in-cell NMR structure determination in insect Sf9 cells from the available chemical shift list[30]. For GB1 at 10 μM concentration only, torsion angle restraints derived with TALOS-N were used in addition to the NOE restrains. The 20 conformers with lowest target function values were subjected to restrained energy refinement with OPALp[57] using the AMBER force field[58]. The structures at 50 and 10 μM concentration have been deposited in the Protein Data Bank with accession codes 7QTR and 7QTS, respectively. The chemical shift assignments have been deposited in the BioMagResBank (BMRB) database with accession code 34700.

**Statistics and reproducibility**. The results are expressed in bar plots that show the corresponding means + SD. $p$ values lower that 0.05 obtained with one-way analysis of variance (ANOVA), followed by Dunnett's post hoc test were considered significantly different of controls. In all cases replicates are biologically independent samples. Sample size and number of replicates was determined based on previous studies.

**Reporting summary**. Further information on research design is available in the Nature Portfolio Reporting Summary linked to this article.

## Data availability

All data are available in the main text or the supplementary materials. Source data of all plots can be found in Supplementary Data 1. All the plasmids generated in this work will be available through materials transfer agreements. NMR data and protein structures have been deposited in public databases (BMRB 34700, 34701; PDB 7QTR, 7QTS).

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

## Acknowledgements

We are grateful to Bernd Roschitzki from the Functional Genomics Centre Zurich for access to mass spectrometry instrumentation and to the Scientific Center for Optical and Electron Microscopy (ScopeM) of ETH Zurich. JAG, RR acknowledge the Swiss National Science Foundation for the grant Sinergia 154461. JAG, RR acknowledge the Swiss National Science Foundation for the grant: Sinergia CRSII5_177195. JAG and NCP acknowledge the Wilhelm Hurka Stiftung grant. PG acknowledges the grant-in-Aid for Scientific Research of the Japan Society for the Promotion of Science (20 K06508). JAG, RR acknowledge the grant Synapsis Stiftung Switzerland. M.B. acknowledges the Krebsliga grant: KFS-4903-08-2019.

## Author contributions

Conceptualization: J.A.G., N.C.P., H.K., P.G., R.R. Methodology: J.A.G., N.C.P., H.K., D.G., M.B. Investigation: J.A.G., N.C.P., P.G., R.R. Visualization: J.A.G., N.C.P., H.K., P.G., R.R. Supervision: J.A.G., R.R. Writing—original draft: J.A.G., N.C.P., P.G., R.R. Writing—review & editing: J.A.G., N.C.P., H.K., P.G., R.R.

## Competing interests

The authors declare no competing interests.
