## [Peer Review File · Communications Biology]

Reviewers' comments:

Reviewer #1 (Remarks to the Author):

In this manuscript the authors describe a compelling approach to delivering isotopically labeled proteins into mammalian cells with the objective of determining the NMR structure of a protein in an intracellular environment. The ability to study the structure of proteins inside of cells is very important conceptually and is of value to many fields. The authors present a detailed and technical description of their approach to evaluate the reporter system for transexpression, advancing in cell NMR, and determining the structure of GB1 in mammalian cells. The majority of this manuscript is outside of my main area of technical expertise in proteomics, so I can only comment on the small piece of this work where the authors appropriately deployed PRM in Figure 3. That being said, I enjoyed reading this manuscript and found it to be written in such a way that it was approachable to an outsider to NMR. In addition, the transexpression system developed here is interesting to consider for non NMR based analyses. Given these limitations of my technical ability to fully review this manuscript, I do not have any recommended revisions and find the work appropriate for publication in Nature Communications.

Reviewer #2 (Remarks to the Author):

The manuscript by Gerez et al reports on the improvements in in-cell NMR technology to study protein structures at atomic resolution in live cells. The authors developed a new reporter that allowed them to optimize "transexpression" of exogenous proteins in various cell lines and solve the structure of GB1 to 1.1 Å accuracy in the protein backbone. The work is very extensive and some aspects of the methodology are novel. I think that the term "transexpression" is very appropriate. I have a few concerns:

1. It is not clear why the described reporter method is better than transexpression of eGFP into cells. An increase in fluorescence is an easy readout of this system. One can also directly transexpress luciferase. In this case, one can directly determine the success of transexpression by using luminol.
2. The authors should comment on the reasons why different transexpression protocols result in vastly different outcomes.
3. In Fig.1 it is not clear why cytosolic eGFP is mostly present in the nucleus.
4. In Fig.2 the quality of the confocal images is very low. It is not possible to assess the co-localization of the transexpressed proteins at this resolution.
5. Fig.3A images are very puzzling. The in-buffer and in-cell peaks have very similar linewidths. It is difficult to reconcile these spectra with the well-understood fact that the cellular viscosity is at least 3 times higher than that of buffers.

Reviewer #3 (Remarks to the Author):

Highly efficient delivery of isotope labeled protein into mammalian cells is vital for the success of in-cell NMR, the emerging technique to obtain structural or dynamical information of proteins at atomic resolution in living cells. Juan Gerez and co-authors optimized electroporation (EP) protocol using their developed reporter system and demonstrated their protocol in several disordered and structural proteins. I have severe concerns about the originality and novelty of the results.

- 1) Similar studies have been performed in Selekno group. The deliver efficiency by CPP, SLO, EP methods were systematically quantified, the optimized EP protocol and cell type dependent efficiency were also described in Beata Bekei dissertation (the dissertation can be download at <https://refubium.fu-berlin.de/handle/fub188/4508>). Although the authors claim they used new reporter system, there are not many advantages compared to that by directly using fluorescence labeled proteins.
- 2) The authors did not provide or explain the critical parameters for high efficiency delivery in their

protocol. Buffer, electroporator device, pulse program, protein concentration, which is more critical and why?

3) Besides delivery efficiency, the challenge for denovo structural determination of proteins in cells is resonance assignments due to the low sensitivity of 3D experiments. The authors need to show they can do the assignments in cells if they claim the highly efficiency "transexpression" method can be used to solve the structure in mammalian cells.

Reviewer #1 (Remarks to the Author):

In this manuscript the authors describe a compelling approach to delivering isotopically labeled proteins into mammalian cells with the objective of determining the NMR structure of a protein in an intracellular environment. The ability to study the structure of proteins inside of cells is very important conceptually and is of value to many fields. The authors present a detailed and technical description of their approach to evaluate the reporter system for transexpression, advancing in cell NMR, and determining the structure of GB1 in mammalian cells. The majority of this manuscript is outside of my main area of technical expertise in proteomics, so I can only comment on the small piece of this work where the authors appropriately deployed PRM in Figure 3. That being said, I enjoyed reading this manuscript and found it to be written in such a way that it was approachable to an outsider to NMR. In addition, the transexpression system developed here is interesting to consider for non NMR based analyses. Given these limitations of my technical ability to fully review this manuscript, I do not have any recommended revisions and find the work appropriate for publication in Nature Communications.

Reviewer #2 (Remarks to the Author):

The manuscript by Gerez et al reports on the improvements in in-cell NMR technology to study protein structures at atomic resolution in live cells. The authors developed a new reporter that allowed them to optimize "transexpression" of exogenous proteins in various cell lines and solve the structure of GB1 to 1.1 Å accuracy in the protein backbone. The work is very extensive and some aspects of the methodology are novel. I think that the term "transexpression" is very appropriate. I have a few concerns:

1. It is not clear why the described reporter method is better than transexpression of eGFP into cells. An increase in fluorescence is an easy readout of this system. One can also directly transexpress luciferase. In this case, one can directly determine the success of transexpression by using luminol.

We thank the reviewer for this comment. We are aware and familiar with transexpressing fluorescent and fluorescent-labeled proteins and in principle this is another possibility. We decided to develop a reporter system because there are a few advantages over fluorescent proteins to evaluate transexpression as outlined in the following:

First, in our reporter the transexpressed Gal4-VP16 needs to reach the cell nucleus to activate the reporter gene for eGFP. This is an advantage as the method of choice assures that the transexpressed protein actually reaches its final destination within the cell and is functional.

Second, it is known that many proteins have affinity for lipids and membranes. Many of these proteins can remain attached to the cells but from the extracellular domain. We have seen this behavior with recombinant eGFP where a substantial amount of protein gets attached to the cells even when the electroporation pulse is intentionally not applied.

Last, in addition to protein delivery, our reporter also allows to test whether or not the transexpressed protein remains functional (and structurally intact) upon delivery.

We have added a small paragraph explaining the advantages of our reporter system in the main text and discussion of the manuscript.

2. The authors should comment on the reasons why different transexpression protocols result in vastly different outcomes.

We have added this comment in the main text and discussion. In addition to that, we have added a new figure that shows the transexpression levels and cell viability of two cell lines and five protocols.

Based in our experience, the main differences between the protocols in terms of (electroporation-based) transexpression efficiency are the buffers and devices used for electroporation. While the Nucleofector IIb (AMAXA) yields good transexpression efficiency with buffers such as “R”, the Neon (Invitrogen) is superior when PBS is used. These differences might be due to the fact that the two devices use different electroporation units (cuvettes versus tips)

Instrument:	Nucleofector IIb		Nucleofector IIb		Nucleofector IIb		NEON		NEON	
	Settings: Q-001		Settings: Q-001		Settings: A-001		Settings: 1700mV 200mSec		Settings: 1700mV 200mSec	
Buffer:	"R"		PBS		"R"		"R"		PBS	
	T	V	T	V	T	V	T	V	T	V
Cos7	++	++	+	+	+	+	++	+	++++	+++
A2780	+++	++	+	+	+	+	+++	+	+++++	+++

Figure Sx. Summary of five representative transexpression protocols used in this work. Two cell lines (Cos7 and A2780) were tested using electroporator devices of two manufacturers were used in this work Nucleofector-IIb (AMAXA-LONZA) and Neon (Invitrogen). The settings for the AMAXA electroporator were fixed to protocols Q-001 and A-001, while for the Invitrogen electroporator were 1700mV and 200mSec. The buffers used were buffer “R” (AMAXA-LONZA) and PBS. Transexpression (denoted with a “T”) and cell viability (denoted with a “V”) were assayed with the rGAL4-VP16 reporter system and the trypan blue test, respectively.

3. In Fig.1 it is not clear why cytosolic eGFP is mostly present in the nucleus.

We and others have seen this behavior in the past in different cell lines and models that do not rely in transexpression. For instance, in Fig S3A and S3G in Cos7 and neuron-like cells of a previous publication of our group (Sci Transl Med . 2019 Jun 5;11(495):eaau6722. doi: 10.1126/scitranslmed.aau6722.). We have observed a reduction of nuclear staining at very low levels of eGFP expression. The explanation of the nuclear localization of eGFP can be found in more systematic analyses (Anal Biochem . 2007 Sep 1;368(1):95-9. doi: 10.1016/j.ab.2007.05.025.)

4. In Fig.2 the quality of the confocal images is very low. It is not possible to assess the co-localization of the transexpressed proteins at this resolution.

We are sorry for the low quality of these images. The image quality very likely was lost upon conversion of the original image to pdf in the merged document. A new pdf file was made.

5. Fig.3A images are very puzzling. The in-buffer and in-cell peaks have very similar linewidths. It is difficult to reconcile these spectra with the well-understood fact that the cellular viscosity is at least 3 times higher than that of buffers.

We thank the reviewer for this comment. The two spectra show very similar cross peak forms because the in-cell samples contain significantly more amount of protein than the one in buffer and the signal-to-noise was adjusted to illustrate the superposition of the spectra. At similar protein concentrations it is evident that the in-cell spectra show substantial line broadening compared to the one in buffer. This is now explained and shown in a new supplementary figure (see below).

Last but not least, the in-cell effect on NMR resonances is explained in a another manuscript from our group that is under revision and contains a detailed study on the line broadening resonances and other effects in cells

Figure Sx. In-cell NMR of GB3. Two dimensional $[^{15}\text{N},^1\text{H}]$ -HMQC NMR experiments of ^{15}N -labeled GB3, in buffer (black spectrum) and mammalian cells (red spectrum). This experiment shows spectra obtained with approximately the same concentration of GB3 in both buffer and cells. The overlap is shown on the right. NMR measurements were carried out at 37 °C.

Reviewer #3 (Remarks to the Author):

Highly efficient delivery of isotope labeled protein into mammalian cells is vital for the success of in-cell NMR, the emerging technique to obtain structural or dynamical information of proteins at atomic resolution in living cells. Juan Gerez and co-authors optimized electroporation (EP) protocol using their developed reporter system and demonstrated their protocol in several disordered and structural proteins. I have severe concerns about the originality and novelty of the results.

1) Similar studies have been performed in Selekno group. The deliver efficiency by CPP, SLO, EP methods were systematically quantified, the optimized EP protocol and cell type dependent efficiency were also described in Beata Bekei dissertation (the dissertation can be download at <https://refubium.fu-berlin.de/handle/fub188/4508>). Although the authors claim they used new reporter system, there are not many advantages compared to that by directly using fluorescence labeled proteins.

We have transexpressed fluorescent and fluorescent-labeled proteins in the past. Our reporter system has few advantages compared to using fluorescent proteins to evaluate transexpression. These advantages are crucial for structural analyses at physiological conditions as explained below.

First, in our reporter the transexpressed Gal4-VP16 needs to reach the cell nucleus to activate the reporter gene for eGFP. This is an advantage as the method of choice assures that the transexpressed protein actually reaches its final destination within the cell and is functional.

Second, it is known that many proteins have affinity for lipids and membranes. Many of these proteins can remain attached to the cells but from the extracellular domain. We have seen this behavior with recombinant eGFP where a substantial amount of protein gets attached to the cells even when the electroporation pulse is intentionally not applied.

Last, in addition to protein delivery, our reporter also allows to test whether or not the transexpressed protein remains functional (and structurally intact) upon delivery.

We have added a small paragraph explaining the advantages of our reporter system in the main text and discussion of the manuscript.

2) The authors did not provide or explain the critical parameters for high efficiency delivery in their protocol. Buffer, electroporator device, pulse program, protein concentration, which is more critical and why?

We have added this information in the main text and discussion. In addition to that, we have added a new figure that shows the transexpression levels and cell viability of two cell lines and five protocols.

Based in our experience, the main differences between the protocols in terms of (electroporation-based) transexpression efficiency are the buffers and devices used for electroporation. While the Nucleofector IIb (AMAXA) yields good transexpression efficiency with buffers such as “R”, the Neon (Invitrogen) is superior when PBS is used. These differences might be due to the fact that the two devices use different electroporation units (cuvettes versus tips)

Instrument:	Nucleofector IIb		Nucleofector IIb		Nucleofector IIb		NEON		NEON	
	Settings:		Settings:		Settings:		Settings:		Settings:	
Buffer:	"R"		PBS		"R"		"R"		PBS	
	T	V	T	V	T	V	T	V	T	V
Cos7	++	++	+	+	+	+	++	+	++++	+++
A2780	+++	++	+	+	+	+	+++	+	+++++	+++

Figure Sx. Summary of five representative transexpression protocols used in this work. Two cell lines (Cos7 and A2780) were tested using electroporator devices of two manufacturers were used in this workNucleofector-IIb (AMAXA-LONZA) and Neon (Invitrogen). The settings for the AMAXA electroporator were fixed to protocols Q-001 and A-001, while for the Invitrogen electroporator were 1700mV and 200mSec. The buffers used were buffer “R” (AMAXA-LONZA) and PBS. Transexpression (denoted with a “T”) and cell viability (denoted with a “V”) were assayed with the rGAL4-VP16 reporter system and the trypan blue test, respectively.

3) Besides delivery efficiency, the challenge for denovo structural determination of proteins in cells is resonance assignments due to the low sensitivity of 3D experiments. The authors need to show they can do the assignments in cells if they claim the highly efficiency “transexpression” method can be used to solve the structure in mammalian cells.

From our point of view there are two approaches for structure determination in cells by NMR. In one approach a de novo collection of data is obtained including triple resonance experiments such as shown in Figures 4 and S4 which is sufficient for backbone assignment of GB1. In another approach, in vitro data, deposited structures or/and assignments are used and adjusted to the in-cell NMR spectra demonstrated here. The latter approach can be related to the very successful molecular replacement strategy in x-ray crystallography. Furthermore, we are currently developing machine learning techniques that request only one HNCA, 2D HSQCs and a 3D $^{13}\text{C},^{15}\text{N}$ -resolved combined $^1\text{H},^1\text{H}$ -NOESY spectra for fully automated structure determination with a success rate of 75% for proteins with a molecular weight of up to 20 kDa (tested on 100 proteins) indicating that the bottle neck is no longer on the triple resonance experiments.

REVIEWERS' COMMENTS:

Reviewer #1 (Remarks to the Author):

The revised manuscript has been significantly improved and my comment have been properly addressed.

Reviewer #2 (Remarks to the Author):

The revised manuscript is improved and addressed my concerns.